# Quantum interference of currents in an atomtronic SQUID

C. Ryu [1✉], E. C. Samson [1,2] & M. G. Boshier [1]

Quantum interference of currents is the most important and well known quantum phenomenon in a conventional superconducting quantum interference device (SQUID). Here, we report the observation of quantum interference of currents in an atomtronic SQUID. Analogous to a conventional SQUID, currents flowing through two junctions in an atomtronic SQUID interfere due to the phase difference from rotation. This interference results in modulation of critical currents. This modulation was observed for three different radii with clear modulation periods which were measured to be consistent with fundamental rotation rates. This observation shows the possibility of studying various interesting SQUID physics with an atomtronic SQUID and especially, macroscopic quantum phenomena with currents may be realized with an atomtronic SQUID toward the goal of quantum metrology of rotation sensing.

---

[1] Physics Division, Los Alamos National Laboratory, Los Alamos, NM, USA. [2] Department of Physics, Miami University, Oxford, OH, USA. ✉email: cryu@lanl.gov

A superfluid in a loop with Josephson junctions has been studied for the past several decades because of its unique quantum phenomena and applications in quantum sensing[1] and information processing[2]. One of the most important quantum phenomena in such a system is the quantum interference of currents, after which superconducting quantum interference devices (SQUIDs) were named. In a conventional SQUID, the electrons in superconductors are subject to phase twists due to the external magnetic field and the periodic modulation of critical currents that results from quantum interference led to the development of the direct current (DC) SQUID as one of the most sensitive magnetometers[3,4]. Using neutral atoms of superfluid helium, it has been shown that the phase twist induced by the physical rotation of the device creates quantum interference of currents, making rotation sensing possible[5,6]. Furthermore, the atomtronic SQUID[7,8], an atomtronic[9] analog of a SQUID that uses a Bose–Einstein condensate (BEC), has been developed to explore quantum phenomena of a SQUID with a dilute quantum gas.

An atomtronic SQUID consists of a toroidal trap and tunneling barriers that act as Josephson junctions. Similar to the conventional SQUID, an atomtronic SQUID can be made with a different number of Josephson junctions for different applications. With a single junction atomtronic SQUID, phase slips[7] and hysteresis[10] have been observed, showing similarity to a radio frequency (RF) SQUID. With a double junction atomtronic SQUID, Josephson effects[8] and resistive flow[11] have been demonstrated, but the quantum interference of currents has not been observed until now.

Here we report the first observation of quantum interference of currents in an atomtronic SQUID through the measurement of the periodic modulation of the critical current. Analogous to a DC SQUID, the phase twist from external rotation produces periodic modulation of the critical current due to quantum interference. We observe periodic modulation of a critical atom number, which is equivalent to modulation of the critical current, for three different atomtronic SQUID radii that were chosen to demonstrate the variation of the modulation periods. Theoretically, the modulation period is identical to the rotation rate of atoms circulating within the loop with a unit winding number, which may be called the fundamental rotation rate $\Omega_0$[12]. The measured modulation periods are consistent with the directly measured $\Omega_0$, confirming that the observed periodic modulation was the result of rotation-induced quantum interference. This realization of a DC atomtronic SQUID makes it possible to create a sensitive and compact rotation sensor. In addition, many intriguing quantum phenomena of the conventional SQUID can be studied with a dilute quantum gas. In particular, the quantum state of currents may be manipulated by controlling the physical rotation, enabling the creation of macroscopic quantum states for many interesting applications, including quantum metrology of rotation sensing[13–16] and quantum information processing[17].

## Results

**DC SQUID physics.** The periodic modulation of the critical current can be understood by calculating the total current within a model of the atomtronic SQUID based on quantum phase-controlled Josephson junction currents and a toroidal trap geometry (Fig. 1a). The total current is the result of quantum interference of the two Josephson junction currents, given by

$$I_1 = \frac{1}{2}\left(I_t + I_j\right) = I_c\sin\emptyset_1 \qquad (1)$$

and

$$I_2 = \frac{1}{2}\left(I_t - I_j\right) = I_c\sin\emptyset_2, \qquad (2)$$

where $I_c$ is the critical current of atoms, $I_t$ is the total current, and $I_j$ is the circulating current around the atomtronic SQUID. Because of the toroidal geometry and single valuedness of the wavefunction describing the atoms, the phases should satisfy

$$\emptyset_1 - \emptyset_2 + 2\pi\omega = 2\pi n, \qquad (3)$$

where $\omega = \frac{\Omega}{\Omega_0}$, with $\Omega$ being the rotation rate of atoms, and $n$ is an integer. The rotation rate of the atoms can be shown to be

$$\omega = \omega_{ext} + \beta_{atom}\frac{I_j}{I_c}, \qquad (4)$$

where $\omega_{ext} = \frac{\Omega_{ext}}{\Omega_0}$, $\Omega_{ext}$ is the external rotation rate of the atomtronic SQUID, $\beta_{atom} = \frac{2\pi I_c}{N\Omega_0}$, and $N$ is the total number of atoms. This equation for the rotation rate of atoms can be derived from the relation between the circulating current and the movement of atoms relative to the Josephson junctions. The parameter $\beta_{atom}$ is analogous to the screening parameter in the conventional SQUID and can be thought as proportional to the inductance, which induces the deviation of the rotation rate of atoms from the imposed external rotation rate of the atomtronic SQUID.

Equations (1)–(4) are equivalent to those of a DC SQUID[4], reflecting the fact that the fundamental underlying physics of a double junction atomtronic SQUID and a DC SQUID is the same. In the limit of $\beta_{atom} = 0$ (for example, when $I_c \approx 0$ with much higher barrier height), we can analytically calculate the total currents: $I_t = 2I_c\cos(\pi\omega_{ext})\sin(\emptyset_1 - \pi\omega_{ext})$. Thus the critical current is $|2I_c\cos(\pi\omega_{ext})|$, which establishes a clear modulation of the critical currents with a period of $\Omega_0$. With finite $\beta_{atom}$, we can numerically calculate the critical current, and the periodic modulation amplitude decreases with the increasing $\beta_{atom}$, as can be seen in Fig. 1b.

By using the calculated modulation in Fig. 1b, the expected periodic modulation of the critical current in an atomtronic SQUID was calculated with the Gross–Pitaevskii equation (GPE)[8,18] in two dimensions (2D). Figure 1c shows the normalized critical current (see "Methods" for the details on the Josephson effects in an atomtronic SQUID), which is the critical current of atoms normalized to the number of atoms ($2I_c/N$), as a function of the number of atoms for the different rotation rates of the atomtronic SQUID. For a fixed number of atoms, the normalized critical current shown in Fig. 1c modulates with rotation rate. However, it is very difficult to experimentally observe this modulation because of the strong dependence of the normalized critical current on the number of atoms and the difficulty in producing a BEC with the same number of atoms consistently. Instead of a fixed number of atoms, we therefore used a fixed normalized bias current, generated by moving Josephson junctions with a fixed speed. When the rotation rate changes, the critical atom number—which is the number of atoms at the transition from DC to alternating current (AC) Josephson effect with the chosen normalized bias current—modulates periodically, as shown in the GPE calculation of Fig. 1d. We chose this modulation of the critical atom number as a way to observe the quantum interference of currents.

**Realization of a DC atomtronic SQUID.** In the previous work[19], painted potentials (see "Methods" on painted potentials for details), time-averaged potentials generated by scanning laser beams, were used to create various atomtronic circuits[20], and this method was used here to create a double junction atomtronic

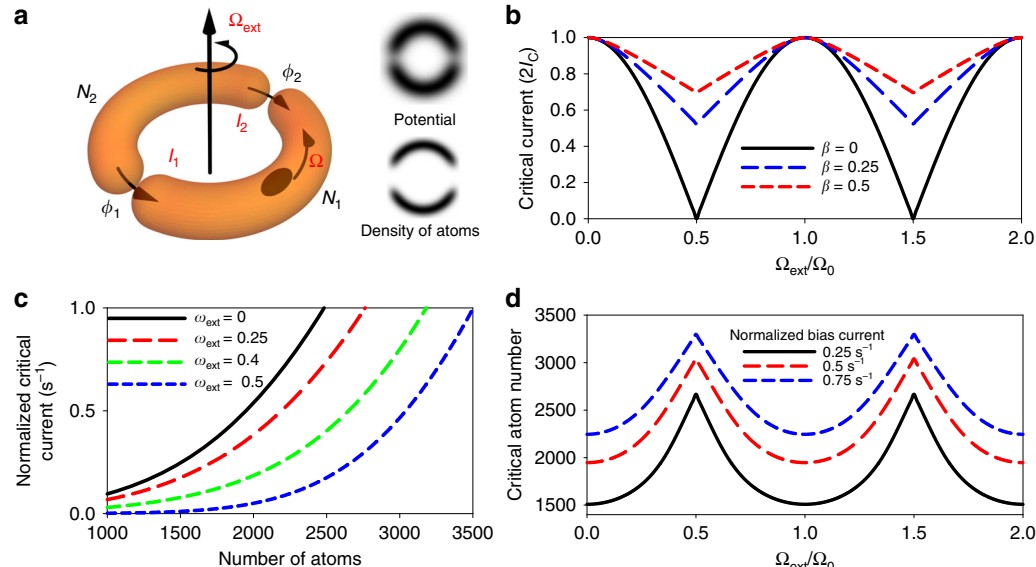

**Fig. 1 Calculation of the periodic modulation of the critical current. a** Schematic of a double junction atomtronic SQUID. The atomtronic SQUID was created by scanning a single 834 nm laser beam with 1.7 μm waist and the barrier full width at half maximum (FWHM) was 2.1 μm. $\Omega_{ext}$ is the rotation rate of the atomtronic SQUID and $\Omega$ is the rotation rate of atoms. $\varnothing_1$ and $\varnothing_2$ are the phase differences across the Josephson junctions, $I_1$ and $I_2$ are Josephson junction currents, and $N_1$ and $N_2$ are numbers of atoms in each half. Arrows represent the movement of the junctions. The calculated potential of the atomtronic SQUID and the density of atoms are shown for the radius of 3.85 μm. **b** Critical current as a function of $\Omega_{ext}/\Omega_0$ calculated for different values of $\beta_{atom}$. **c** Normalized critical currents ($2I_c/N$) where $I_c$ is the critical current and $N$ is the total number of atoms as a function of the number of atoms with different $\omega_{ext}$ for the atomtronic SQUID with 3.85 μm radius. $\beta_{atom}$ varies with the number of atoms and the critical current. For each number of atoms, $\beta_{atom}$ was calculated to find the variation of the normalized critical current. **d** Modulation of the critical atom number as a function of $\Omega_{ext}/\Omega_0$ for 3 different normalized bias currents with the 3.85 μm radius atomtronic SQUID.

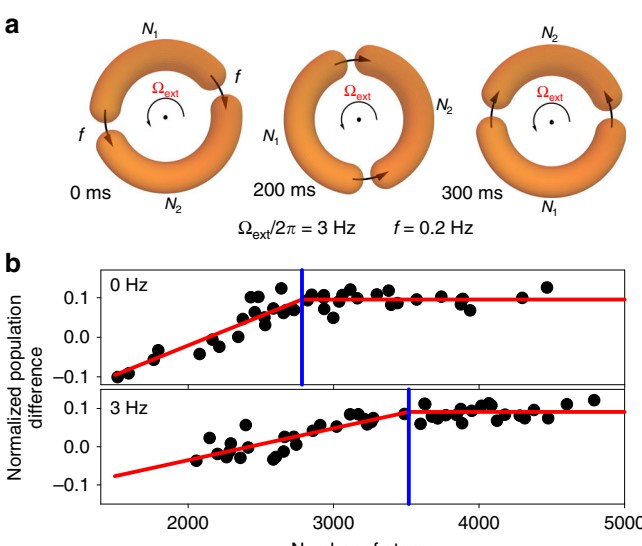

**Fig. 2 Movement of Josephson junctions for the measurement of the critical atom number. a** Movement of Josephson junctions during the experimental observation of quantum interference. The total duration of the junction movement used to create a bias current was 300 ms with a 3 Hz rotation frequency ($\Omega_{ext}/2\pi$) and a 0.2 Hz frequency for the bias current generation ($f$). **b** Variation of $z$ as a function of the number of atoms for two different rotation frequencies with fitting to determine the critical atom number for the atomtronic SQUID of radius 4.82 μm. The red line shows the best fit with a piecewise function of a constant and a straight line with a slope with the blue line indicating the transition point and critical atom number.

SQUID, as shown in Fig. 1a. The change in the height, thickness, and location of the barriers in the atomtronic SQUID was controlled by lowering the power of the laser beam at the specific locations during the scanning.

The $^{87}$Rb BEC was created in a double junction atomtronic SQUID with varying rotation rates (see "Methods" on the BEC production for details). Then two junctions were moved to each other to create a normalized bias current of 0.8 s$^{-1}$ as shown in Fig. 2a, and rotation rates for two junctions were chosen to be $\Omega_{ext} + 2\pi f$ and $\Omega_{ext} - 2\pi f$ to make this possible. After this junction movement step, the number of atoms was measured via in situ absorption imaging to determine the normalized population difference $z = \frac{N_2 - N_1}{N}$ (see Fig. 1a). The change of $z$ from DC (constant $z$ due to the tunneling of atoms maintaining same density across the junctions and $N > N_c$, where $N_c$ is the critical atom number) to AC (decreasing $z$ due to the decreasing of the net tunneling of atoms resulting in the density difference across the junctions and $N < N_c$) Josephson effect can be seen in Fig. 2b for two different rotation rates of the atomtronic SQUID. To determine the critical atom number, the data were fitted with a piecewise function of a straight line with a slope and a constant with a transition point between the two as a fitting parameter. A clear shift in the critical atom number can be seen for the two different rotation rates, as shown in Fig. 2b. Figure 3a–c shows the change in the measured critical atom number as a function of rotation rates of the atomtronic SQUID for three different radii. All three data sets show clear, periodic modulation. The modulation periods were determined by fitting the data sets to a damped sine function. As mentioned earlier, this modulation may be used to sense rotation. With 4.82-μm radius atomtronic SQUID, from the slope of the modulation curve and atom-shot-noise-limited atom

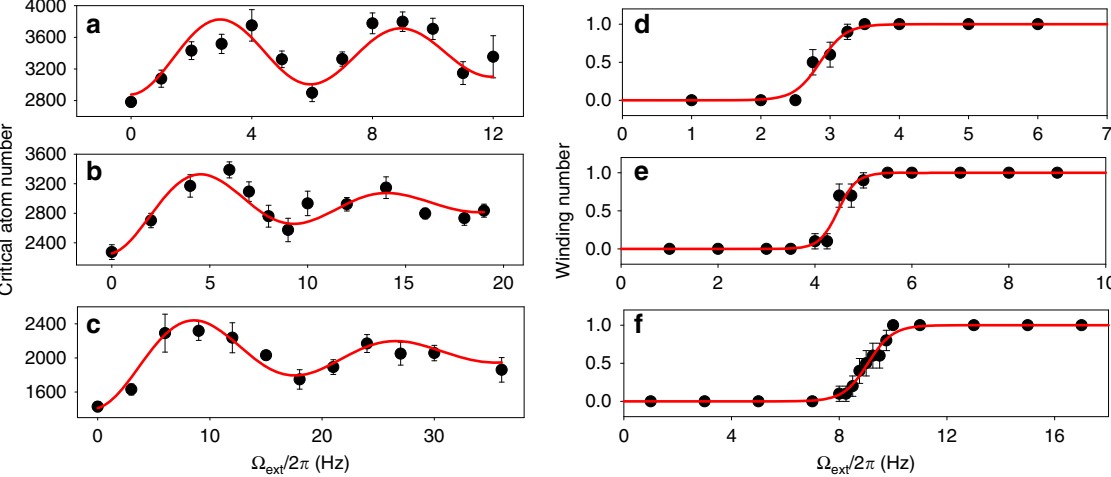

**Fig. 3 Measurement of the periodic modulation of the critical atom number.** The error bars in the plots represent the standard error and the numbers inside the brackets represent one standard deviation uncertainty. The uncertainty in the radius of the atomtronic SQUID consists of the systematic uncertainty in the estimation of the radius, which is 3% of the radius and the statistical uncertainty. The radius for **a** and **d** is 4.82 $(0.14)_{sys}$ $(0.01)_{stat}$ μm, for **b** and **e** is 3.85 $(0.11)_{sys}$ $(0.02)_{stat}$ μm, for **c** and **f** is 2.891 $(0.084)_{sys}$ $(0.022)_{stat}$ μm. The modulation periods for **a–c** were measured with a damped sine function fitting. The transition point from the 0 to 1 winding number for **d–f** was determined with the fitting function $1/(1 + e^{-k(\Omega_{ext} - \frac{\Omega_0}{2})})$. Modulation periods (Hz) were found to be 5.99 (0.22), 9.52 (0.48), and 17.99 (0.64) for **a–c**, respectively. $\Omega_0/2\pi$ (Hz) were measured to be 5.70 (0.10), 8.96 (0.12), and 18.11 (0.35) for **d–f**, respectively.

number sensitivity of 50, the sensitivity of 0.1 Hz over 1 min integration time may be achieved. The sensitivity can be improved by increasing the radius $r$, as it scales as $1/r^2$. Further improvements can be made by employing entanglement techniques from quantum metrology[21].

In order to compare $\Omega_0$ with the measured modulation periods, $\Omega_0$ must be directly measured. With a rotating system, minimization of free energy shows that, when $n - \frac{1}{2} < \frac{\Omega_{ext}}{\Omega_0} < n + \frac{1}{2}$, where $n$ is an integer, the winding number should be equal to $n$. Since this transition involves an abrupt change in the winding number from 0 to 1 at $\Omega_{ext} = \frac{\Omega_0}{2}$, determining this transition point is the most accurate way of measuring $\Omega_0$. To measure $\Omega_0$, during the evaporation a single Josephson junction was rotated at a constant rate. After the end of evaporation, absorption images were taken to determine the winding number[22] (see "Methods" on the winding number determination for details). Figure 3d–f shows the results of this measurement for the same three radii that were used previously for the modulation measurement. All plots show a distinct transition of the winding number from 0 to 1, as expected, and $\Omega_0$ was measured from this transition. The measured $\Omega_0$ values are in good agreement with the measured modulation periods, as can be seen from Fig. 3, confirming that the observed periodic modulation is due to quantum interference of currents from the rotation-induced phase twist.

**Comparison with GPE calculations.** Calculating $\Omega_0$ is non-trivial since the width of the wavefunction of atoms is significant compared to the radius of the trap[23]. The calculation was performed by minimizing the energy of a BEC in a rotating frame with the GPE in 2D. Figure 4a shows a comparison of the measured and calculated values of $\Omega_0$. The best fit value for the calibration scale for the radius estimation based on the $\Omega_0$ measurement data is 0.96, which lies just outside the one-sigma uncertainty range of calibration scale (0.97 and 1.03). One possible reason for this small shift is the change in the value of $\Omega_0$ from small imperfections in the trap potential. This and other

possible mechanisms for the shift may be studied further in the future.

The modulation waveforms of data sets in Fig. 3 show a few differences compared to the theoretical curve in Fig. 1d. To study these differences in more detail, we performed a dynamic 2D GPE simulation of the experimental sequence for the measurement of the critical atom number. Figure 4b–d shows the comparison for the data sets and DC SQUID theory curves. There is a good agreement in the waveform shape between the GPE simulation and DC SQUID theory. However, there is a constant shift of the critical atom number between these two calculations. We believe that this shift occurs because the dynamics is not completely adiabatic. Thereby, small residual phase fluctuations initiate a transition from the DC to the AC Josephson effect, with the net effect of reducing the critical current, and consequently increasing the critical atom number. With the GPE simulations, there is still a difference compared to the experimental data. The main difference is a decrease in the oscillation amplitude of the second oscillation. This damping-like effect may be caused by static perturbations of the potential and non-rotating thermal atoms. A dedicated study with varying amplitude of the static perturbations and thermal atoms might further elucidate this effect in the future.

## Discussion

The good agreement between the DC SQUID theory and GPE simulation shows that this system is ideal for realizing various quantum phenomena of the conventional SQUID with a dilute quantum gas. For example, the atomtronic SQUID offers the possibility to study macroscopic quantum effects by utilizing its ability to detect various many body states with high resolution and sensitivity[24]. With this unique capability, it may be possible to simulate and study many quantum phenomena of the conventional SQUID to solve various urgent open questions regarding the nature of macroscopic quantum states[25]. The other promising direction lies in creating macroscopic superpositions of distinct angular momentum states. These states may be used for quantum metrology of rotation sensing, quantum information

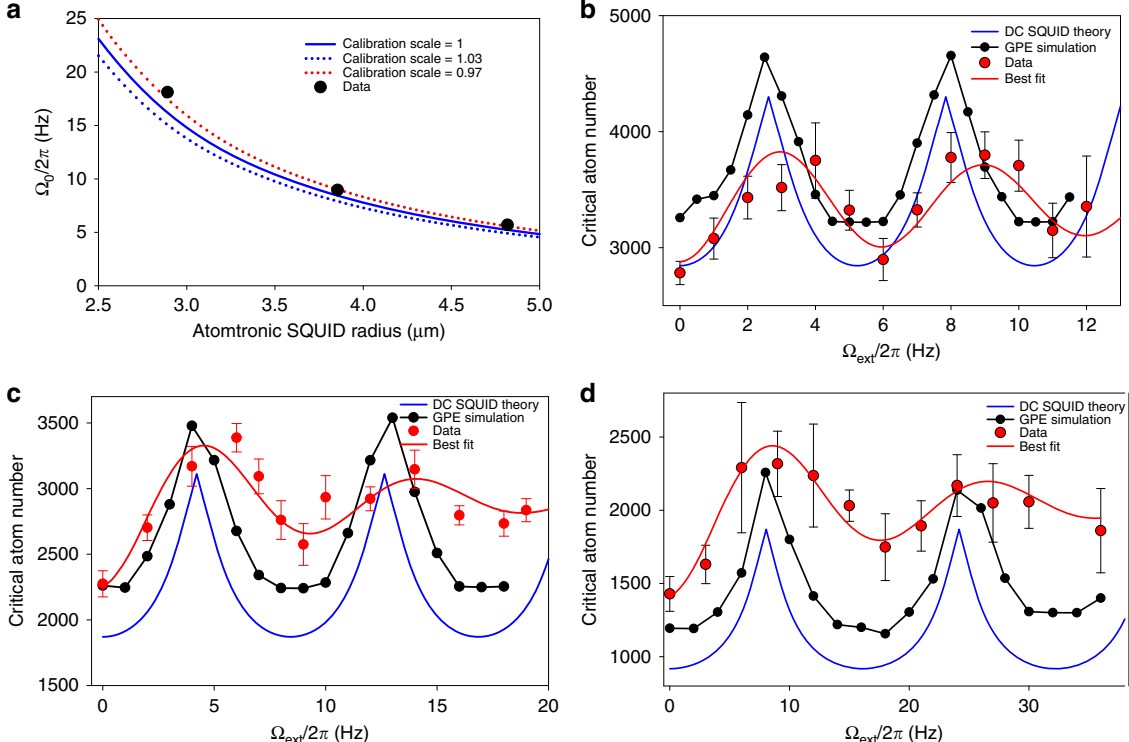

**Fig. 4 Comparison between experiment and theory.** The error bars in the plots represent the standard error. **a** Comparison of the measured and calculated values of $\Omega_0/2\pi$ (calculation done using GPE). The three curves correspond to calibration scales of the atomtronic SQUID radius. **b-d** Critical atom number as a function of the rotation rates obtained with GPE simulation and DC SQUID theory, along with the measured data and the best fit. For **b**, the radius is 4.82 μm; for **c**, the radius is 3.85 μm; and for **d**, the radius is 2.891 μm.

processing, and a possible test of macroscopic realism, along with the Leggett–Garg inequality[26].

## Methods

**Josephson effects in an atomtronic SQUID.** In the previous work[8], Josephson effects were demonstrated with an atomtronic SQUID by measuring the critical current through the comparison with the bias current. It was shown that the normalized current defined as $\dot{z} = 2I/N$, where $I$ is the current of the atoms, $N$ is the total number of atoms, and $z = \frac{N_2 - N_1}{N}$ is the normalized population difference, is a useful way to describe the Josephson effect[18] of an atomtronic SQUID. By moving two junctions close to each other with a rotation frequency $f$, a bias current can be induced. The normalized bias current can be calculated to be $\dot{z}_0 = 4f$, where $z_0$ is the equilibrium normalized population difference for equal chemical potential and density of atoms. The Josephson effect is in a DC regime when $\dot{z}_c > \dot{z}_0$ and in an AC regime when $\dot{z}_c < \dot{z}_0$, where $\dot{z}_c$ is the normalized critical current and there is an atom density difference between two regions of the atomtronic SQUID in the AC regime.

**Generation of painted potentials.** Painted potentials are time-averaged optical potentials from fast scanning laser beams[19]. Two painting beams were used for this experiment. One was a vertical painting beam with a wavelength of 834 nm and waist of 1.7 μm, and the other was a horizontal painting beam with a wavelength of 1064 nm and waist of 12 μm. The painting frequency was 15 kHz for the vertical beam and 33 kHz for the horizontal beam. The vertical beam painted the atomtronic SQUID with a trap depth of 82 nK, a barrier full width at half maximum of 2.1 μm, a barrier height of 42 nK, a radial trap frequency of 520 Hz, and a radius varying from 2.89 to 4.82 μm. The horizontal beam painted a line to create a flat potential to trap atoms against gravity. This created a box-shaped horizontal trap. This horizontal beam also creates a vertical trap. The width of this trap was 39 μm with a trap depth of 1.3 μK and a vertical trap frequency of 297 Hz. The painting was performed using an acousto-optic modulator (AOM) by modulating the frequency and amplitude of the input to the AOM using arbitrary waveform generators.

**Creation of a BEC in an atomtronic SQUID.** The $^{87}$Rb atoms from a magneto optical trap were transferred to the quadrupole magnetic trap. RF evaporation cooling was used to cool atoms in the quadrupole trap down to 10 μK. The transfer of atoms into the optical trap was done adiabatically to minimize heating. The optical trap consisted of a vertical painted potential of the rotating atomtronic SQUID and a horizontal line potential. During evaporation, the vertical trap

remained constant with a fixed rotation rate forming a dimple trap on top of the much deeper horizontal trap that has a weak axial confinement. The horizontal trap depth was reduced from 40 μK to final depth of 1.3 μK to lower the temperature of atoms for the production of the BEC with no discernable thermal atoms. After the BEC was produced, the Josephson junctions were moved to induce a bias current. To make this junction movement as adiabatic as possible, the frequency of the junction movement increased from 0 to 0.2 Hz for 200 ms and then maintained at 0.2 Hz for the next 100 ms. The total movement of the junction ranged from −7.2 to 7.2 degrees where 0 degree corresponded to the symmetric configuration of Josephson junctions in an atomtronic SQUID.

**Measurement of the winding number states of atoms.** The atomtronic SQUID potential was turned off, and after 12.7 ms, absorption images of atoms were taken to measure the winding number. After this long expansion, the momentum distribution of atoms was being measured, and when the winding number $n$ was equal to zero, there was a peak at the center of images of atoms because atoms had zero angular momentum in the trap. With a nonzero $n$ and corresponding angular momentum, there was a hole at the center of images of atoms and the size of the hole was proportional to $n$ and the angular momentum of atoms. The data were fitted with Gaussian distributions in order to determine the winding number. For the experiments reported here, the winding number was either 0 or 1.

## Data availability

The data that support the findings of this study are available from the corresponding authors upon reasonable request.

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

## Acknowledgements

This work was supported by the U.S. Department of Energy through the LANL-LDRD program.

## Author contributions

All three authors participated in the planning of the experiment, data taking, and analysis and writing of the manuscript.

## Competing interests

The authors declare no competing interests.
