## [Peer Review File · Nature Communications]

Reviewers' comments:

Reviewer #1 (Remarks to the Author):

Report on 'Quantum Interference of Currents in an Atomtronic SQUID'

In their manuscript C. Ryu et al describe experiments with two Josephson junctions in a circular atom trap and study evidence for quantum interference of currents in such an quantum gas SQUID through the measurement of the periodic modulation of the critical atom number.

The measured modulation periods were consistent with the directly measured Ω_0 , confirming that the observed periodic modulation was the result of rotation induced quantum interference.

This is an interesting experiment, that potentially warrants publication, but there are many loose ends which need to be considered.

The whole discussion of the quantum gas DC squid in the initial paragraphs relies on a perfect system, with no phase fluctuations, and at zero temperature. What would be the role of fluctuations and of temperature in this discussion.

The drawings in fig 1 and 2 suggest a well-defined quasi 1d path for the atoms between the two Josephson Junctions. How would the real potential look like? It would be good to have it shown (at least in the suppl. material) together with the GP calculation of the atoms in the potential. With the radius not much larger than the width of the painting beam I imagine that the trap depth of 82nK is towards that outside, and not towards the center. Will that have an effect?

It would be good to explain the physical interpretation of what is β_{atom} .

Figure 1 needs a much improved caption ...

in (b) you show critical currents for different β . Which β did you use in (c) or (d)?

Figure 1c: what is the normalized critical current? Normalized to what?, what is ω ? And much more

Fig4b for which radius was this calculation (I guess the is the middle one). Why not show all 3 ring sizes.

The authors compare with GP, why did they not try stochastic GP or Truncated Wigner to include temperature? The calculations are in full 3d?

In the abstract the authors elude on possible application as a rotation sensor. They should give an estimate on how good their sensor currently is ... and what to expect. Are they competitive with atom interferometers?

The authors should give more details on how the experiment is done, how the traps are formed and how they are loaded and most importantly they need to give more information on their experimental parameters. Points that need to be answered / improved:

* I do not understand the way the BEC is produced. The vertical trap is only 82 nK deep, and there is a horizontal trap that is given to be 1.3 μ K deep. I guess during the loading it is much deeper and there is also vertical confinement added to close the optical trap capable to load 10 μ K atoms into.

* The notion of vertical trap frequency for the 'horizontal' trap is very confusing.

* What is the temperature of the atoms?

* What is the chemical potential

* What is the plasma frequency of the employed Josephson Junctions?

* What is normalized bias current?

...

Reviewer #2 (Remarks to the Author):

The manuscript describes an experimental study of a ring-shaped atomic Bose-Einstein condensate forming atomtronic analog of a SQUID. The results show clear evidence of the Josephson current interference and its dependence on the rotation rate correlates with the prediction of DC SQUID equations. This is definitely a novel result which will be highly appreciated by the atomtronics community. Therefore I would happily support these results to be accepted for publication. However many explanations in the text are insufficiently specific and sometimes even confusing, so I think it should be carefully revised. Here are some specific points where I believe the manuscript can be improved:

1. The paragraph in lines 47-58 describes the main equation for atomic DC SQUID. It would be useful to have some more explanation of physical meaning of these equations and various quantities introduced, e.g. I_s and β_{atom} are the ones that are not explained anyhow. Also line 59 starts with "These equations ..." It is not really clear which equations are referred to. Probably main equations should be formatted as numbered formulas and referred to accordingly.
2. Both rotation frequencies and frequency ratios are denoted with a letter omega and termed as "rotation rate" which is confusing. Maybe authors can use different notation for those quantities to improve readability.
3. It is said that some external rotation is imposed on atomic SQUID. However there is no information on how this is done in practice. I can only guess that this is done by additional rotation of the barriers, so that two barriers move with frequencies $\Omega_{\text{ext}+f}$ and $\Omega_{\text{ext}-f}$. Is that correct? In any case this has to be properly described in the text.
4. Most of the experimental results in the manuscript are presented in terms of the critical atom number. It is said that this quantity is equivalent to the Josephson critical current. It is however not shown how these quantities are related. Also in the discussion of how N_c is determined (Fig.2b) it will be useful to explain why is constant Z expected for DC Josephson regime and growing Z for AC.
5. The experimental results in Fig.3 are fitted with a damped sine function. Is there a physical reason for that? The curves from theoretical models show quite different behavior.
6. The role of the ring radius is not elaborated in the text. Is there some physical explanation of why the damping is higher for a smaller ring?

Sincerely,
Yuriy Bidasjuk

Reviewer #3 (Remarks to the Author):

The manuscript by Ryu, et. al., describes the first realization of a dc-SQUID analogue in a atomtronic-like system. The data is technically sound, the paper's conclusions are correct given the data presented, the result is novel, and the result is an important demonstration of an effect long predicted in these ring-type Bose-Einstein condensate systems. As such, it represents an important result in the emerging field of atomtronics, and, more broadly, will be of interest of many BEC experimentalists and theorists. On the other hand, it is quite unclear whether this

system or similar atomtronic systems can be used for the advertised sensitive rotation sensor or testing quantum realism and the like, limiting its reach beyond the aforementioned BEC community. Nevertheless, I believe the paper is of sufficient importance that it should be published in Nature Communications, after addressing the following concerns.

My first concern is the lack of dissipation in the authors' models, both by using zero-temperature GPE and models of the Josephson junctions that exclude any resistive component. The damped interference fringes observed in the present experiment, which are not reproduced by the zero-temperature GPE, would suggest that dissipation plays an important role. While microscopically this dissipation mechanism would be some complicated interaction between the zero-temperature GPE wavefunction and the surrounding thermal cloud or the thermally excited phonons, it may be possible to capture the essential physics using an analog from the superconducting world. Ref. [16], for example, achieved reasonable agreement using a modified RSCJ model for their weak links. I would be interested that if the authors used a similar model for their system, if it would reproduce the damping observed in the interference fringes with only the addition of one or two more free parameters. Furthermore, if an agreement is found, the value of those parameters in relevant units might lend further insight into the physical behavior of the system.

My second concern relates to the labeling of currents. There are currents labeled I_1 and I_2 in figure 1, but those labels are not used in the text. Including them as $I_t = I_1 + I_2$ when introducing the total current might be useful to the reader. Perhaps more useful would be to label $I_1 = I_t/2 + I_S$ and $I_2 = I_t/2 - I_S$, where I_S is the circulating current will make the factor of 2 that appears in the equation for I_S obvious. I would also avoid labeling the circulating current as I_S , as this is typically reserved for supercurrent in the resistive models described above. Along the same lines, what does the "f" label mean next to the junctions in Fig. 2?

Finally, I have few other minor comments:

1. Do the authors have a comment on the importance of tunneling barriers to making this device work. Other, similar experiments used weak links. Do those groups have no hope of seeing this effect?
2. At line 139, the authors advertise that the device here has "a very small decoherence rate", which is not measured and, I do not see any basis upon which the authors can make this claim. If the damping of the interference fringes is any indication, the decoherence properties of the current system are not "very small".
3. Ref [14] has a typo in the year.

Comments and questions from reviewer 1

In their manuscript C. Ryu et al describe experiments with two Josephson junctions in a circular atom trap and study evidence for quantum interference of currents in such a quantum gas SQUID through the measurement of the periodic modulation of the critical atom number.

The measured modulation periods were consistent with the directly measured Ω_0 , confirming that the observed periodic modulation was the result of rotation induced quantum interference.

This is an interesting experiment, that potentially warrants publication, but there are many loose ends which need to be considered.

Q1

The whole discussion of the quantum gas DC squid in the initial paragraphs relies on a perfect system, with no phase fluctuations, and at zero temperature. What would be the role of fluctuations and of temperature in this discussion.

The relative phase fluctuation of atoms connected through a Josephson junction is a very important subject with many previous studies. For a double well system with small phase limit, it can be shown that the quantum mechanical phase fluctuation can be written as (PRL 96, 130404 (2006)).

$$\langle \phi^2 \rangle \approx \sqrt{E_c/4E_j}$$

Where E_c is the capacitance energy from interaction between atoms and E_j is the Josephson coupling energy. For the typical conditions in the experiment, $\langle \phi^2 \rangle \approx 0.001 \sim 0.004$, showing that the quantum mechanical phase fluctuation is negligible.

For the thermally induced phase fluctuation, the important parameter is $k_b T/E_j$. If this value is much smaller than 1, the phase fluctuation is negligible. In our experiment, no discernible thermal atoms were detected, making it difficult to determine the temperature of the system. But the chemical potential of atoms can be calculated to be about 22nK~42nK in the typical conditions. The temperature may be estimated to be 1/2 of the chemical potential from the typical evaporation dynamics. Then the temperature is around 11nK~21nK. With the typical conditions, $k_b T/E_j \approx 0.3 \sim 0.009$. Where the large value comes from the conditions where number of atoms is smaller. This shows that thermally induced phase fluctuation is not very significant.

Based on this estimation, the discussion of the quantum gas DC SQUID can rely on the system without quantum and thermal phase fluctuations.

Q2

The drawings in fig 1 and 2 suggest a well-defined quasi 1d path for the atoms between the two Josephson Junctions. How would the real potential look like? It would be good to have it shown (at least in the suppl. material) together with the GP calculation of the atoms in the

potential. With the radius not much larger than the width of the painting beam I imagine that the trap depth of 82nK is towards that outside, and not towards the center. Will that have an effect?

The calculated potential of the atomtronic SQUID and the density of atoms were added to the Fig.1 which is changed in the revised manuscript.

As you pointed out, the actual minimum position of the atomtronic SQUID potential is different from the painting radius due to the finite size of the painting beam. From the overlap of the potential, the actual radius is smaller than the painting radius. For example, for 3.85 μm radius with 1.7 μm waist, the actual radius of the potential is 3.75 μm .

Q3

It would be good to explain the physical interpretation of what is beta_atom.

The equation of $\omega = \omega_{\text{ext}} + \beta_{\text{atom}} \frac{I_j}{I_c}$ (4) describes the difference between the rotation rates of atoms and the atomtronic SQUID. β_{atom} determines magnitude of this difference. In the conventional SQUID, this is called screening parameter which determines the magnetic flux difference from the external magnetic flux and is proportional to inductance. Analogously, with the atomtronic SQUID, β_{atom} can be thought as proportional to the “inductance” like term which induces the deviation of the rotation rate of atoms from the imposed external rotation rate of the atomtronic SQUID. For example, when the barrier height is much higher than the chemical potential, the critical current is almost zero with $\beta_{\text{atom}} \approx 0$. In this case, the rotation rate of atoms and the atomtronic SQUID will be the same. This discussion is included in the revised manuscript (line 76-78 and 81).

Q4

Figure 1 needs a much improved caption ...

in (b) you show critical currents for different beta. Which beta did you use in (c) or (d)?

Figure 1c: what is the normalized critical current? Normalized to what?, what is small omega? And much more

The parameter β_{atom} is defined as $\beta_{\text{atom}} = \frac{2\pi I_c}{N\Omega_0}$. From this definition, it can be seen that β_{atom} varies with number of atoms and critical current. For **c** and **d**, since the number of atoms and critical current varies β_{atom} varies accordingly. For example, for **c** if we fixed number of atoms, critical current is fixed and β_{atom} is fixed too. The normalized critical current is the critical current of atoms normalized to the number of atoms which can be written as $2I_c/N$. Small omega is defined as $\omega = \frac{\Omega}{\Omega_0}$. The caption for Fig.1 is revised based upon these (line 304-311).

Q5

Fig4b for which radius was this calculation (I guess the is the middle one). Why not show all 3 ring sizes.

The radius for Fig.4b is 3.85 μm .

Fig.4 now has comparison between theories and data sets for all three radii. The revised Fig.4 is included in the revised manuscript.

Q6

The authors compare with GP, why did they not try stochastic GP or Truncated Wigner to include temperature? The calculations are in full 3d?

The GP calculations are done with 2D simulation since atoms are confined in the uniform vertical trap with no dynamics in vertical direction. We confirmed that the full 3D calculation agrees with the 2D calculation. This is clarified in the revised manuscript (line 87-88, line 146, and line 154).

The incorporation of temperature related effect into GPE simulation with stochastic GPE or Truncated Wigner is a good suggestion. But we need to develop this capability for the full simulation. For the timely publication of the current manuscript, the simulation with stochastic GPE or Truncated Wigner may be presented in the future paper with the next phase of experiments where we can change the temperature of the system to study the thermal effects more systematically.

Q7

In the abstract the authors elude on possible application as a rotation sensor. They should give an estimate on how good their sensor currently is ... and what to expect. Are they competitive with atom interferometers?

The rotation sensing sensitivity depends on the radius of the atomtronic SQUID. To make this competitive with atom interferometers, the radius has to be much larger, which is in principle possible but practically challenging. With 4.82 micron radius, the variation of 800 in the critical number of atoms corresponds to 2.8Hz change. If we can measure the critical number of atoms with the sensitivity of 10. The rotation sensing sensitivity is 0.035Hz. With 482 micron radius, the sensitivity can be improved by 10^4 which will be 3.5×10^{-6} Hz. With the current conditions, this method of rotation sensing is not competitive with an atom interferometer. However, this is a completely novel way of rotation sensing and with further improvement in increasing the size of the atomtronic SQUID and possible application of quantum metrology, the future atomtronic SQUID may have a comparable rotation sensing capability. A brief summary of this consideration is included in the revised manuscript (line 124-129).

Q8

The authors should give more details on how the experiment is done, how the traps are formed and how they are loaded and most importantly they need to give more information on their experimental parameters. Points that need to be answered / improved:

**** I do not understand the way the BEC is produced. The vertical trap is only 82 nK deep, and there is a horizontal trap that is given to be 1.3 μK deep. I guess during the loading it is much deeper and there is also vertical confinement added to close the optical trap capable to load 10 μK atoms into.***

The optical trap consisted of a vertical painted potential of the rotating atomtronic SQUID and a horizontal painted line potential. During evaporation, the vertical trap remained constant forming a dimple trap on top of the much deeper horizontal trap which has a weak axial confinement in addition to the radial confinement. The horizontal trap depth was reduced from 40 μK to 1.3 μK to lower the temperature of atoms. During this process, the atoms outside the vertical dimple trap were removed from the trap and at the end of the evaporation process, a BEC was formed inside the rotating atomtronic SQUID. This description is added to the method section of the revised manuscript (line 207-210).

** The notion of vertical trap frequency for the 'horizontal' trap is very confusing.*

The horizontal beam painted a line to create a flat potential to trap atoms against gravity. This created a box-shaped horizontal trap. This horizontal beam has a symmetric waist of 12 μm , resulting in a vertical trap. The trap frequency of this vertical trap from the horizontal beam is called "vertical trap frequency".

** What is the temperature of the atoms?*

As mentioned earlier, during this experiment, we lowered the trap depth to create a BEC with no discernible thermal atoms and reliable measurement of temperature was not possible with imaging of atoms. But we can estimate the maximum temperature from the calculated value of chemical potentials.

The chemical potential is in the range of 22nK~42nK. If we assume the temperature of 1/2 of the chemical potential, the maximum temperature estimation may be in the range of 11nK~21nK.

** What is the chemical potential*

As mentioned in the previous answer, the chemical potential range is 22nK~42nK.

** What is the plasma frequency of the employed Josephson Junctions?*

The plasma frequency of the Josephson Junctions varies depending on the number of atoms. In typical conditions, the range of the plasma frequency is 3Hz~37Hz.

** What is normalized bias current?*

The normalized bias current is the bias current of atoms normalized to the number of atoms.

Comments and questions from reviewer 2

The manuscript describes an experimental study of a ring-shaped atomic Bose-Einstein condensate forming atomtronic analog of a SQUID. The results show clear evidence of the Josephson current interference and its dependence on the rotation rate correlates with the prediction of DC SQUID equations. This is definitely a novel result which will be highly appreciated by the atomtronics community. Therefore I would happily support these results to be accepted for publication. However many explanations in the text are insufficiently specific and sometimes even confusing, so I think it should be carefully revised. Here are some specific points where I believe the manuscript can be improved:

Q1

The paragraph in lines 47-58 describes the main equation for atomic DC SQUID. It would be useful to have some more explanation of physical meaning of these equations and various quantities introduced, e.g. I_s and β_{atom} are the ones that are not explained anyhow.

These equations describe the relationship between Josephson currents and phases in the regime of DC Josephson effect with no chemical potential difference. The first and second equation simply state that the current flowing in each Josephson junction can be written as combination of the total current and circulating current. The third equation states that the phase difference of two Josephson Junctions should satisfy the condition of the single valuedness of the wave function. The fourth equation states that the difference between the external imposed rotation rate and the rotation rate of atoms is proportional to the β_{atom} and circulating current I_j which is the current circulating inside the atomtronic SQUID. As discussed in the earlier answer, the parameter β_{atom} is equivalent to the screening parameter in the conventional SQUID. This introduces the deviation of the rotation rate of atoms from the imposed external rotation rate of the atomtronic SQUID.

Also line 59 starts with "These equations ..." It is not really clear which equations are referred to. Probably main equations should be formatted as numbered formulas and referred to accordingly.

In the revised manuscript, the equations are numbered and referred to them with equation numbers (line 64, 65, 69, and 72).

Q2

Both rotation frequencies and frequency ratios are denoted with a letter omega and termed as "rotation rate" which is confusing. Maybe authors can use different notation for those quantities to improve readability.

Throughout the revised manuscript, clear distinctions is made to distinguish rotation frequencies and frequency ratios. In the manuscript, rotation rate means rad/s and rotation frequency means Hz. This distinction is also clearly made in the revised manuscript.

Q3

It is said that some external rotation is imposed on atomic SQUID. However there is no information on how this is done in practice. I can only guess that this is done by additional rotation of the barriers, so that two barriers move with frequencies $|\Omega_{\text{ext}}+f$ and $|\Omega_{\text{ext}}-f$. Is that correct? In any case this has to be properly described in the text.

Your description of the movement of junctions is correct. There is overall rotation of the atomtronic SQUID with rotation rate of Ω_{ext} and the relative movement of the junction into each other with the rotation frequency of f . To make this happen, the rotation rate of the junction 1 was $\Omega_{\text{ext}} + 2\pi f$ and the rotation rate of the junction 2 was $\Omega_{\text{ext}} - 2\pi f$ as you described. This is described in the revised manuscript (line 110-111).

Q4

Most of the experimental results in the manuscript are presented in terms of the critical atom number. It is said that this quantity is equivalent to the Josephson critical current. It is however not shown how these quantities are related. Also in the discussion of how N_c is determined (Fig.2b) it will be useful to explain why is constant Z expected for DC Josephson regime and growing Z for AC.

The critical atom number is the number of atoms when the critical current is equal to the bias current. As mentioned in the manuscript and from the Fig.1d, with a fixed number of atoms, the critical current variation with the rotation rate can be measured but due to the fluctuation in the number of atoms, this measurement is not practical. With fixed normalized bias current, the change in the critical current can be measured through the measurement of the variation of the critical atom number. This shows that same physical mechanism of the quantum interference of currents can be measured either with critical current variation or critical atom number variation.

With the DC Josephson effect, tunneling of atoms maintain the same density across the junction, resulting in that z is not changed from the geometrically determined value. However, with the AC Josephson effect, the net tunneling of atoms decrease, resulting in the density difference across the junction with the accompanying change in z . This explanation is added in the revised manuscript (line 113-117).

Q5

The experimental results in Fig.3 are fitted with a damped sine function. Is there a physical reason for that? The curves from theoretical models show quite different behavior.

We did the fitting with a damped sine function to find the frequency and damping without any assumption of the underlying theory. As discussed in the manuscript, the deviation from the DC SQUID theory results may come from static perturbations of the potential and non-rotating thermal atoms. A dedicated future study will make it possible to find out the complete model for the data.

Q6

The role of the ring radius is not elaborated in the text. Is there some physical explanation of why the damping is higher for a smaller ring?

Although we don't have a complete answer yet, one possibility is the increase of the fundamental rotation rate of the atomtronic SQUID which is proportional to $1/r^2$. Because of this increase of the fundamental rotation rate in the smaller radius atomtronic SQUID, there will be bigger difference in the velocity between the stationary defects and atoms and the rotating atoms and Josephson junctions in the atomtronic SQUID. This may produce bigger damping like effect. The careful study of this effect in experiments and theories will be done in the future.

Comments and questions from reviewer 3

The manuscript by Ryu, et. al., describes the first realization of a dc-SQUID analogue in a atomtronic-like system. The data is technically sound, the paper's conclusions are correct given the data presented, the result is novel, and the result is an important demonstration of an effect long predicted in these ring-type Bose-Einstein condensate systems. As such, it represents an important result in the emerging field of atomtronics, and, more broadly, will be of interest of many BEC experimentalists and theorists. On the other hand, it is quite unclear whether this system or similar atomtronic systems can be used for the advertised sensitive rotation sensor or testing quantum realism and the like, limiting its reach beyond the aforementioned BEC community. Nevertheless, I believe the paper is of sufficient importance that it should be published in Nature Communications, after addressing the following concerns.

Q1

My first concern is the lack of dissipation in the authors' models, both by using zero-temperature GPE and models of the Josephson junctions that exclude any resistive component. The damped interference fringes observed in the present experiment, which are not reproduced by the zero-temperature GPE, would suggest that dissipation plays an important role. While microscopically this dissipation mechanism would be some complicated interaction between the zero-temperature GPE wavefunction and the surrounding thermal cloud or the thermally excited phonons, it may be possible to capture the essential physics using an analog from the superconducting world. Ref. [16], for example, achieved reasonable agreement using a modified RSCJ model for their weak links. I would be interested that if the authors used a similar model for their system, if it would reproduce the damping observed in the interference fringes with only the addition of one or two more free parameters. Furthermore, if an agreement is found, the value of those parameters in relevant units might lend further insight into the physical behavior of the system.

This is a very interesting suggestion. Compared to the Ref. [16], the temperature is much lower in our case and the total change in angle is much. This small change in angle was done to minimize the density difference across the Josephson junctions to avoid the effect of dissipative currents. This was confirmed from simulation with 2D GPE, with which we found no evidence of dissipative current with our experimental conditions. RSCJ model itself may not be useful for our case since RSCJ model assumes the periodicity with the magnetic field which is equivalent to the rotation rate for the atomtronic SQUID. The research into the effect of thermal atoms is ongoing and it may be better to separate this work from the current manuscript for the timely publication of this manuscript.

Q2

My second concern relates to the labeling of currents. There are currents labeled I_1 and I_2 in figure 1, but those labels are not used in the text. Including them as $I_t = I_1 + I_2$ when introducing the total current might be useful to the reader. Perhaps more useful would be to label $I_1 = I_t/2 + I_S$ and $I_2 = I_t/2 - I_S$, where I_S is the circulating current will make the factor of 2 that appears in the equation for I_S obvious. I would also avoid labeling the circulating current as I_S , as this is typically reserved for supercurrent in the resistive models described above. Along the same lines, what does the "f" label mean next to the junctions in Fig. 2?

The suggested changes in the notation were made. Now the changed equations are

$$I_1 = I_t/2 + I_j = \sin \phi_1$$

$$I_2 = I_t/2 - I_j = \sin \phi_2$$

Where I_t is the total current and I_j is the circulating current. This change is added in the revised manuscript (line 64-65).

f means the rotation frequency of the junctions for the relative movement to create the bias current.

Q3

Do the authors have a comment on the importance of tunneling barriers to making this device work. Other, similar experiments used weak links. Do those groups have no hope of seeing this effect?

The difference between the tunneling barriers and the weak links is clearer with regard to the sensitivity of the critical current on the number of atoms. Because of the change in the chemical potential as a function of atoms, the critical current changes accordingly. Fig.1c shows this variation. As mentioned in the previous paper (PRL 111 205301 (2013)), with weak links, the slope is much steeper. This means that the variation of the critical atom number is much smaller than with the tunneling barriers. This will require very accurate measurement of the number of atoms to observe the modulation of the critical atom number. This consideration suggests that although in principle, the interference effect can be observed, in practice, the observation of this interference effect with weak links will be very hard.

Q4

At line 139, the authors advertise that the device here has “a very small decoherence rate”, which is not measured and, I do not see any basis upon which the authors can make this claim. If the damping of the interference fringes is any indication, the decoherence properties of the current system are not “very small”.

This is a very good point and the mention of this is removed in the revised manuscript (line 169-171).

Q5

Ref [14] has a typo in the year.

The typo is corrected in the revised manuscript (line 268).

REVIEWER COMMENTS

Reviewer #2 (Remarks to the Author):

I believe, that the updated version of the manuscript can now be accepted for publication.

Reviewer #3 (Remarks to the Author):

I am mostly pleased with the changes to the manuscript. As an aside, the additional discussion about the physical meaning of beta makes me note that the Campbell group measured that quantity directly on the RF-SQUID in PRX 4 031052 (2014). I recommend that the paper be published after addressing the comment below.

I am completely confused by the added discussion of the sensitivity. First, what does it mean to have an "atom number detection sensitivity" of 10? Is that the signal-to-noise in the atom number? Is that what the current experiment is capable of? Is that a factor of 10 improvement over what the current experiment can do? Second, sensitivity only makes sense over a certain integration time. If the experiment could achieve a sensitivity of 35 mHz in 1 μ s of integration time, then in just over 10 s it could measure the rotation of the earth. Of course, I am well aware that the present experiment has a much longer integration time; my example serves only to highlight the importance of this quantity. Finally, I am confused by the final sentence, and in particular the statement "combined with possible application of quantum metrology of rotation sensing"? I assume the authors mean that using quantum entanglement to enhance the measurement, but I think the term "quantum metrology of rotation sensing" is ill-defined. Might I suggest the following revision: "The sensitivity can be improved by increasing the radius r , as it scales as $1/r^2$. Further improvements can be made by employing entanglement techniques from quantum metrology." If my assumption is correct, the authors should add relevant references as well.

Comments and questions from reviewer 2

I believe, that the updated version of the manuscript can now be accepted for publication.

Comments and questions from reviewer 3

I am mostly pleased with the changes to the manuscript. As an aside, the additional discussion about the physical meaning of beta makes me note that the Campbell group measured that quantity directly on the RF-SQUID in PRX 4 031052 (2014). I recommend that the paper be published after addressing the comment below.

Q1

I am completely confused by the added discussion of the sensitivity. First, what does it mean to have an "atom number detection sensitivity" of 10? Is that the signal-to-noise in the atom number? Is that what the current experiment is capable of? Is that a factor of 10 improvement over what the current experiment can do? Second, sensitivity only makes sense over a certain integration time. If the experiment could achieve a sensitivity of 35 mHz in 1 us of integration time, then in just over 10 s it could measure the rotation of the earth. Of course, I am well aware that the present experiment has a much longer integration time; my example serves only to highlight the importance of this quantity.

For the atom number sensitivity, instead of 10, the atom shot noise limited sensitivity of \sqrt{N} is used, where N is the number of atoms. For the number of atoms used for this radius, the atom shot noise limited sensitivity is 50. We did more careful calculation of the estimation. The slope of the interference curve is $\frac{dN}{d\Omega} = 496\text{Hz}^{-1}$ and $\delta N = 50$. With these parameters, the sensitivity of rotation is $\delta\Omega = \frac{\delta N}{\frac{dN}{d\Omega}} \approx 0.1\text{ Hz}$.

This is the sensitivity over one cycle of the experiment which takes about 1 min. Therefore, the rotation sensitivity is 0.1Hz over 1 min of integration time. This change was integrated into the text (line 125-127).

Q2

Finally, I am confused by the final sentence, and in particular the statement "combined with possible application of quantum metrology of rotation sensing"? I assume the authors mean that using quantum entanglement to enhance the measurement, but I think the term "quantum metrology of rotation sensing" is ill-defined. Might I suggest the following revision: "The sensitivity can be improved by increasing the radius r , as it scales as $1/r^2$. Further improvements can be made by employing entanglement techniques from quantum metrology." If my assumption is correct, the authors should add relevant references as well.

Your assumption is correct on the meaning of quantum metrology of rotation sensing. We meant that with quantum entanglement, the measurement sensitivity may be enhanced. The suggested revision was integrated into the text (line 127-129) with the reference of the review paper of “Quantum metrology with nonclassical states of atomic ensembles” (line 129).

REVIEWERS' COMMENTS

Reviewer #3 (Remarks to the Author):

The latest changes are sufficient for the manuscript to be accepted for publication.